# Origin and structure of polar domains in doped molecular crystals

E. Meirzadeh[1,*], I. Azuri[1,*], Y. Qi[2], D. Ehre[1], A.M. Rappe[2], M. Lahav[1], L. Kronik[1] & I. Lubomirsky[1]

Doping is a primary tool for the modification of the properties of materials. Occlusion of guest molecules in crystals generally reduces their symmetry by the creation of polar domains, which engender polarization and pyroelectricity in the doped crystals. Here we describe a molecular-level determination of the structure of such polar domains, as created by low dopant concentrations ($<0.5\%$). The approach comprises crystal engineering and pyroelectric measurements, together with dispersion-corrected density functional theory and classical molecular dynamics calculations of the doped crystals, using neutron diffraction data of the host at different temperatures. This approach is illustrated using centrosymmetric α-glycine crystals doped with minute amounts of different L-amino acids. The experimentally determined pyroelectric coefficients are explained by the structure and polarization calculations, thus providing strong support for the local and global understanding of how different dopants influence the properties of molecular crystals.

[1] Department of Materials and Interfaces, Weizmann Institute of Science, Rehovot 76100, Israel. [2] The Makineni Theoretical Laboratories, Department of Chemistry, University of Pennsylvania, Philadelphia, Pennsylvania 19104-6323, USA. * These authors contributed equally to this work. Correspondence and requests for materials should be addressed to A.M.R. (email: rappe@sas.upenn.edu) or to M.L. (email: meir.lahav@weizmann.ac.il) or to L.K. (email: leeor.kronik@weizmann.ac.il) or to I.L. (email: igor.lubomirsky@weizmann.ac.il).

The presence of small amounts of deliberately introduced additives, commonly called dopants, controls electrical, optical and mechanical properties of many practically important materials[1–3]. In molecular crystals, the incorporation of dopants may introduce a variety of structural distortions to the host. For example, incorporation of tiny amounts of amino acids into one of the most studied ferroelectric molecular crystals, tri-glycine sulfate, strongly affects its electrical and electromechanical properties[1,4]. However, despite the great importance of doping in a wide variety of fields, from pharmacology to large-scale industrial synthesis, the structure of the distorted sites could not be determined at the molecular level due to their small concentrations (often below 0.5%). Studies of crystal doping demonstrated that the formation of a mixed molecular crystal is determined by the structure of the dopants and the structure of surface sites at which the guest molecules bind before their occlusion into the host crystal[5–7]. According to this mechanism, the dopants are incorporated in a polar mode within the bulk of the host[8–10] and thus convert non-polar hosts into mixed polar crystals[9,11–13]. The dopant-induced lattice polarity of the mixed crystal originates from two sources. First, the dopant molecule may have a different dipole moment than that of the host that it replaces. Second, although many non-polar crystals comprise polar molecules but do not have an overall dipole moment since these dipoles cancel each other, the asymmetric distortions introduced by the dopant may force the dipoles of the neighbouring host molecules out of compensation, thereby contributing to the overall polarity of the mixed crystal as well.

Polar crystals develop a surface charge upon temperature variation, since heating and cooling slightly alter the equilibrium positions of the molecules, changing the polarization of the crystal. This phenomenon is known as pyroelectricity[14] and was investigated in detail in a variety of inorganic and molecular crystals, especially in ferroelectric materials[15–17]. Improvements in current measurement equipment during the last decade[18,19] have increased the sensitivity of pyroelectric measurements by at least 10,000 times, allowing pyroelectric coefficient measurements of only $10^{-13}$ C cm$^{-2}$ K$^{-1}$, thereby enabling detailed studies of the polarity of mixed crystals. The pyroelectric effect strongly depends on the structure of the doped polar domains; nevertheless, this structure cannot be directly inferred from the pyroelectric data. Therefore, it is necessary to carry out complementary theoretical computations. These are extremely challenging, due to sheer size of the unit cell with low-concentration dopants, and the complexity of intra- and intermolecular motions in organic crystals. Recent progress in dispersion-corrected density functional theory (DFT) permits inclusion of van der Waals interactions in molecular solids[20,21], where they constitute a very important part of the bonding and are therefore required for reliable calculations.

In the current work, to access the temperature dependence of the polarization within the DFT calculations[15], we include the experimental temperature dependence of the host lattice parameters[22,23] in the calculations. Finally, to assess the effects of anharmonic dynamics, which are computationally inaccessible with DFT for a system of this size, we performed additional classical molecular dynamics (MD) computations. Here we demonstrate the application of this integrated approach in a model system, by determining the equilibrium molecular structure of the polar domains in the centrosymmetric α-glycine crystals doped with different L-α-amino acids. The present approach allows for differentiation between guest molecule contributions and those of the distorted host molecules, thereby providing a way to correlate between the macroscopic polarization and the molecular structure.

## Results

**Crystal engineering.** The α-polymorph of glycine (monoclinic space group $P2_1/n$) contains four molecules per unit cell. The achiral glycine molecules assume chiral conformations in the crystal. Therefore, the crystal can be represented by two pairs of chiral layers (L, L′ and D, D′, Fig. 1) where a $2_1$ symmetry operation transforms L to L′ and D to D′. The site which the guest molecules occupy in the crystal is determined by the layer and attachment energy of the face through which the guest molecules are occluded in the bulk[24]. Consequently, the L-amino acid dopants can be inserted enantioselectively with equal probabilities within the L and L′ layers through the $(0\bar{1}0)$ face of the host (Fig. 1).

**Pyroelectric measurements.** The pure α-glycine crystal is centrosymmetric and therefore not pyroelectric. Growth of these crystals in the presence of L-α-amino acids, for example, alanine, threonine or serine, reduces the symmetry of the host and creates polar domains. Dopants residing in the L and L′ sites induce the same polarization along the $b$ axis, but opposite polarization along the $a$ and $c$ directions of the crystal due to the $2_1$ symmetry parallel to the $b$ axis. Because the zwitterions of the α-amino acids possess a high dipole moment, $\approx 14.9$ D (ref. 25), even a tiny amount, $<0.5\%$ wt wt$^{-1}$, of the dopant results in a detectable pyroelectric effect along the $b$ axis. The magnitude of the pyroelectric current decreases with time after $\tau \approx 7$ ms (Fig. 2a–c), whereas a homogeneous crystal would have produced a constant current in response to a step-like heating from the surface (Supplementary Discussion). This indicates the presence of a concentration gradient as a function of depth, with the surface being the most dopant-enriched. The gradient in dopant concentration can be rationalized by considering the increase of the exposed $(0\bar{1}0)$ face of the growing crystals. This assertion is further supported by high-pressure liquid chromatography measurements, performed on crystalline segments cleaved perpendicular to the polar $b$ axis; the dopant content decreases with depth, and is proportional to the pyroelectric coefficient, $\alpha = \partial P/\partial T$, where $P$ is the polarization and $T$ the

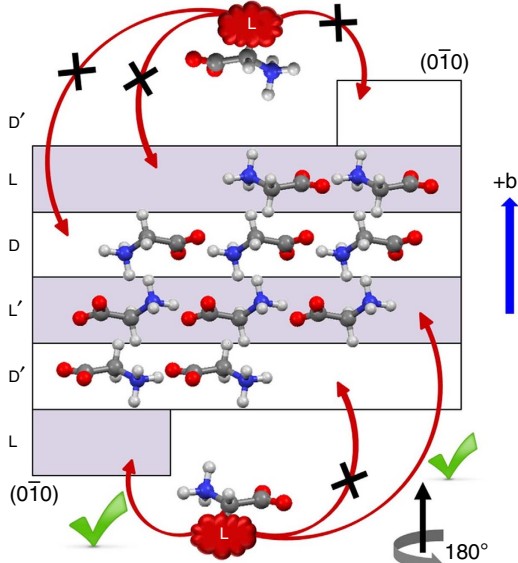

**Figure 1 | Polar occlusion of a guest L-enantiomer through the $(0\bar{1}0)$ face of α-glycine.** The L-amino acids interact enantioselectively with molecules of the D and D′ layers through zwitterionic interactions to occupy the L and L′ sites[5,6].

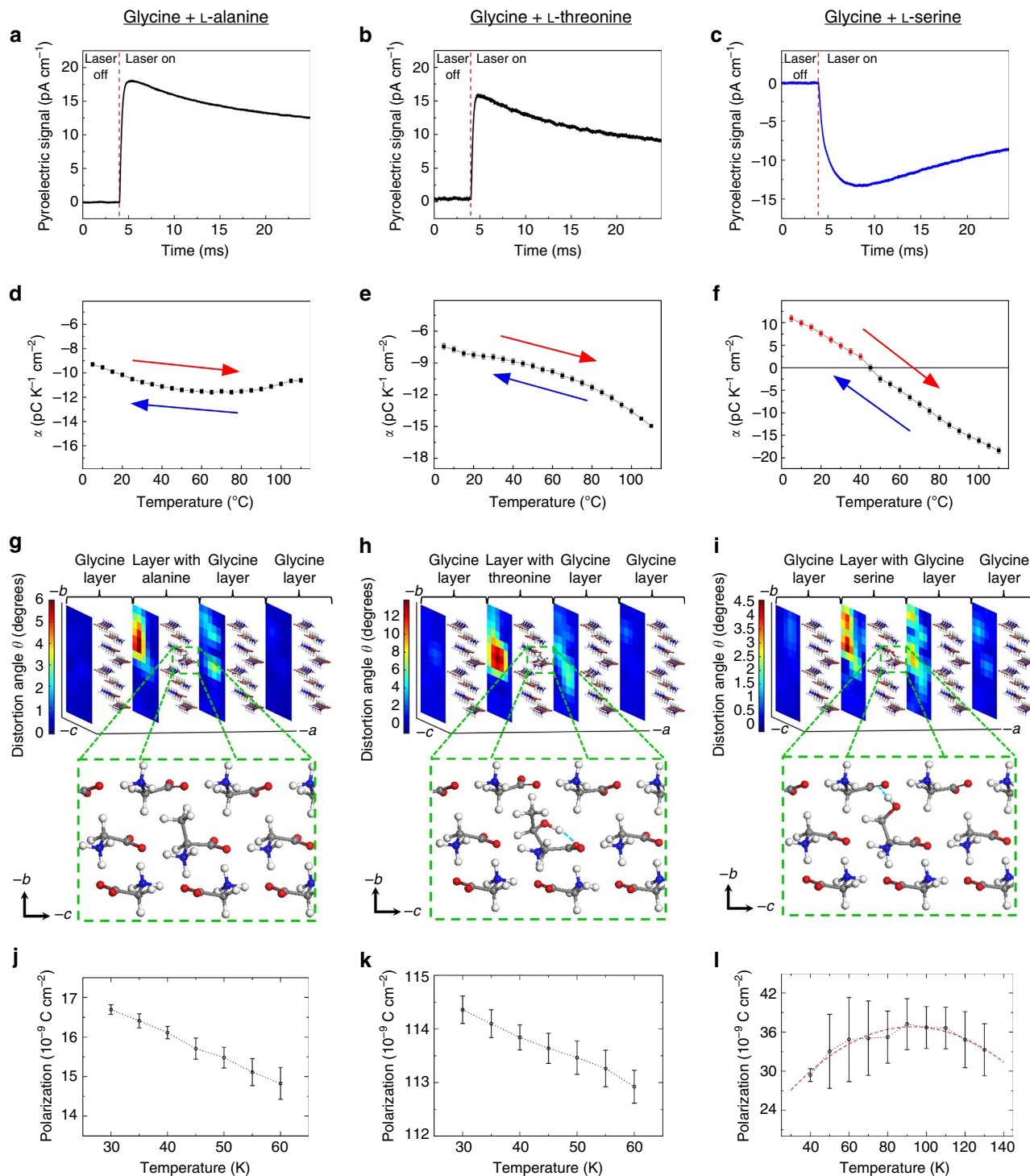

**Figure 2 | Pyroelectricity and crystal structure of α-glycine doped with L-amino acids.** First row (**a–c**): measured pyroelectric signal of the doped crystals at 25 °C. Second row (**d–f**): experimental temperature dependence of the pyroelectric coefficient. Error bars represent s.e.m. values. The pyroelectric effect is fully reversible with temperature and does not decay with time (>10 months), which implies that once occluded, the dopant molecules do not diffuse from L and L′ to the D or D′ sites of the crystal. Third row (**g–i**): DFT computed most stable conformation for each system, along with a three-dimensional intensity map depicting molecular distortion in the unit cell, with the colours representing the dopant-induced distortion angle of the nitrogen to carboxylic carbon vector, relative to its orientation in the undoped glycine crystal (note the different scale for each system). Fourth row (**j–l**): MD-computed temperature-dependent polarization for each system. The dots are average values and the error bars represent the s.d.

temperature (Supplementary Table 1). The thickness of the enriched layer, $d$, can be estimated from the unidirectional thermal diffusion time, $\tau$; $d = \sqrt{2D\tau} \approx 200\,\mu\text{m}$, where $D \approx 0.05\,\text{cm}^2\,\text{s}^{-1}$ is the thermal diffusion coefficient of glycine (Supplementary Discussion).

L-alanine was chosen as a dopant because it is structurally most similar to glycine, where one of its hydrogens is replaced by a methyl group. L-serine and L-threonine are structurally similar between themselves and yet yield dramatically different pyroelectric responses. The pyroelectric coefficient of glycine

doped with alanine or threonine is negative at all temperatures within the range 5–110 °C (Fig. 2d,e), while the pyroelectric coefficient of glycine doped with serine is positive at lower temperatures and becomes negative at higher temperatures (Fig. 2f). This indicates the presence of two sources of polarization with different temperature dependence. As explained above, the occlusion of all L-amino acids takes place via interaction of their similar zwitterionic glycyl groups with the ($0\bar{1}0$) face of the α-glycine crystal (Fig. 1)[26,27] and thus replaces homochiral sites in the crystal. Therefore, the dissimilarity in the pyroelectric response suggests that the difference in the interactions of the side chain of the guest amino acid with the host molecules plays a crucial role in inducing polarity.

**Dispersion-corrected DFT modelling.** To explore the significant, guest-dependent differences in the pyroelectric behaviour and to gain insight into their relation to guest–host interactions, we carried out dispersion-corrected DFT modelling. Initial geometries of the guest molecules were chosen based on their possible open and closed conformations[28–31] that can form a maximum number of hydrogen bonds. Based on this, we found that there is one stable conformation for L-alanine, two low-energy ones for L-threonine, and three accessible conformations for L-serine. The lowest-energy conformation for each system is shown in Fig. 2g–i (the other metastable conformations, including energy differences, are shown in Supplementary Fig. 3). According to the DFT calculations, the guest molecules induce an asymmetric distortion to neighbouring host molecules. We quantify this by considering the distortion angle of the vector pointing from the nitrogen atom to the carboxylic carbon for each molecule, with respect to its value in the unperturbed host. The maximum distortion angle reaches 12°, as observed with the threonine dopant, whereas the smallest distortion is observed with the serine dopant (Fig. 2g–i; for more details, see Supplementary Figs 4–6). Replacing glycine with alanine brings the methyl group of the latter to dislocate just a few neighbouring glycine molecules. Because of the large dipole moment of the glycine host, a distortion of just a few degrees from the original position suffices to induce large polarization. In addition, the deformation is most significant along the *b* direction, because the elastic modulus is smaller along this direction[32]. According to the calculations, threonine and serine exhibit different conformations in the host crystal. In threonine, the hydroxyl hydrogen forms an intramolecular hydrogen bond with one of its own carboxylic oxygens (Fig. 2h). By contrast, serine exhibits a conformation in which the hydroxyl group forms an intermolecular hydrogen bond with an oxygen atom of the carboxylic group of an adjacent deformed glycine molecule (Fig. 2i). The different orientation of the hydroxyl group in threonine and serine provides a first hint for their significantly different temperature-dependence trends of the pyroelectric coefficient.

To explore the pyroelectric trends further, we used the Berry phase method[33] to compute the polarization of glycine doped with L-alanine, L-threonine and L-serine along the measured *b* axis, using DFT. The results, given in the first column of Table 1, show that the total polarization of glycine doped with L-serine is significantly smaller than that of glycine doped with L-alanine or L-threonine. Further insights into this result can be obtained by dividing the total polarization into two distinct contributions. One, given in the second column of Table 1, is the difference between the gas-phase polarization of the guest molecule, but in its geometry within the crystal, and the polarization of the host glycine molecule it substituted, again in the gas-phase but in its ideal geometry within the crystal. This signifies the 'guest contribution' to the overall polarization. The difference between the overall polarization and the 'guest contribution' to it, given in the third column of Table 1, signifies the 'host contribution', namely, the polarization arising from the perturbation of the host molecules by the guest. We readily observe that the three guest molecules behave quite differently in this respect. In L-alanine, almost all of the polarization comes from the guest. In the larger L-threonine, the host is perturbed more significantly and its contribution is dramatically larger. In L-serine, the host is again perturbed significantly, but with polarization of opposite sign. The host polarization is almost exactly equal and opposite to the guest polarization, resulting in a much smaller net polarization. These two opposing dipoles could explain the varying sign of the pyroelectric coefficient in the L-serine-doped glycine by taking into account that the two dipole contributions depend differently on temperature, with one dominating at lower temperatures and the other at higher temperatures.

To calculate different components of the pyroelectric coefficient α, we incorporate key aspects of the *T*-dependent crystal properties. Temperature change influences the polarization in four ways[15]: first, the lattice parameters change with *T*, leaving the internal coordinates (that is, locations of nuclei in the cell) fixed; second, the nuclear coordinates can be relaxed to the *T* = 0 equilibrium positions corresponding to the lattice parameters obtained at the finite *T*; third, the nuclear coordinates can be updated to appropriate finite-*T* equilibrium positions (including effects of anharmonic thermal motions); fourth, the lattice parameters change and 'stretch' the passivating surface charges, thereby changing their areal density and contribution to polarization. Using experimentally determined lattice parameters of the host (from neutron diffraction at a

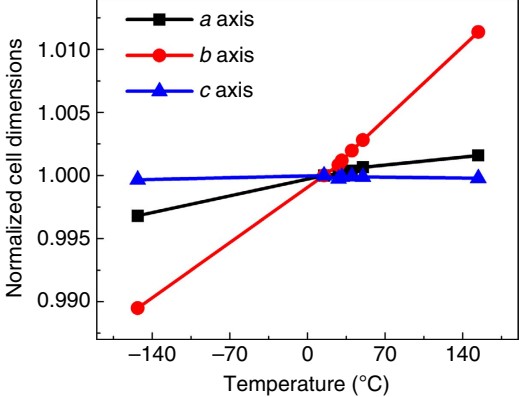

**Figure 3 | Temperature variation of α-glycine cell parameters.** The values are normalized with respect to those at 15 °C (figure is constructed from the data taken from ref. 22). The *b* axis strongly increases with temperature, a far greater change than the changes in the other lattice parameters.

**Table 1 | DFT-computed polarization contributions along the *b* axis.**

| Guest | Total polarization (C cm$^{-2}$) | Guest polarization (C cm$^{-2}$) | Host polarization (C cm$^{-2}$) |
|---|---|---|---|
| L-alanine | $8.6 \times 10^{-9}$ | $9.3 \times 10^{-9}$ | $-0.7 \times 10^{-9}$ |
| L-threonine | $46.2 \times 10^{-9}$ | $15.2 \times 10^{-9}$ | $31.0 \times 10^{-9}$ |
| L-serine | $1.2 \times 10^{-9}$ | $-7.4 \times 10^{-9}$ | $8.6 \times 10^{-9}$ |

All data are normalized to a guest concentration of 0.3%.

range of temperatures, Fig. 3), we incorporate the strain-driven contributions to pyroelectricity.

We can then calculate the average pyroelectric coefficient from a two-point derivative for structures with lattice parameters corresponding to temperatures of 5 and 88 °C, according to the following equation (for more details, see Supplementary Discussion):

$$\alpha \approx \left(\frac{\partial P}{\partial \varepsilon}\right)_{T=0} \left(\frac{\partial \varepsilon}{\partial T}\right)_{\sigma} + \frac{P_0}{S_0}\left(\frac{\partial S}{\partial T}\right)_{\sigma} \qquad (1)$$

where $T$ is the temperature, $P$ is the polarization, $\varepsilon$ is the strain, $\sigma$ is the stress, $S$ is the surface area of the unit cell and the subscript '0' in the second term of the equation refers to values derived from the 5 °C structure. The results obtained ($-3.0 \times 10^{-12}$ C K$^{-1}$cm$^{-2}$, $-8.9 \times 10^{-12}$ C K$^{-1}$cm$^{-2}$ and $-4.8 \times 10^{-12}$ C K$^{-1}$cm$^{-2}$ for alanine-, threonine- and serine-doped structures, respectively) are within a factor of two from experiment for all three dopants, but do not reproduce the unique temperature dependence of the serine-doped crystal. We therefore deduced that the dynamic effect, namely, the variation of polarization due to anharmonicity[15], which is missing from these static DFT calculations, is of importance for distinguishing the pyroelectric trends of the different dopants.

**Molecular dynamics simulations.** To account for dynamic effects, polarization trends have additionally been computed as a function of temperature using classical MD simulations, where the starting structure in an MD simulation was based on the DFT-derived configuration of the dopant and its surroundings. At low temperature, the MD results confirm the DFT-deduced division of polarization between guest and host, as summarized in Table 1 (see Supplementary Table 2 for details). The temperature-dependent polarization, for the three dopants, is given in Fig. 2j–l.

Remarkably, the pyroelectric trends found experimentally in Fig. 2d–f are reproduced by the MD calculation. Specifically, whereas alanine and threonine doping result in a monotonically decreasing polarization, serine doping results in a polarization that increases at low temperature and decreases only at higher temperatures. We note that the critical temperature itself is much lower than the experimental one. This is reasonable, because the force field underlying the MD simulation has not been calibrated for amino acid crystals as a function of temperature, so that full quantitative agreement cannot be expected. Still, the reproduction of the experimental trends informs about the origin of the serine anomaly.

In general, the MD data show that the contribution of molecules at L sites to the pyroelectric response is negative, whereas the contribution of molecules at D sites is positive (Fig. 4). For glycine doped with alanine, the overall pyroelectric response is negative and originates from the change in the polarization of the distorted matrix (Fig. 4a), even though the contribution to the total polarization is mostly from the dopant (Table 1). For glycine doped with threonine, the polarization response of both the dopant site and the distorted host is negative (Fig. 4b). For glycine doped with serine, however, at low temperature, the dopant site (guest-glycine pair) dominates and the overall response is positive (Fig. 4c). At higher temperatures, the effect of temperature on the intermolecular H-bond is reduced, due to the thermal expansion of the crystal along the $b$ axis (Fig. 3), and the negative host response becomes dominant (Fig. 4d). To further verify this mechanism, we have performed additional MD simulations, which considered a higher-energy configuration of serine, which features an intramolecular H-bond, similar to that of threonine (Fig. 2h), rather than an intermolecular H-bond as in Fig. 2i. Indeed, in this case, the polarization was found to decrease monotonically with temperature, as in threonine (see Supplementary Fig. 7 for

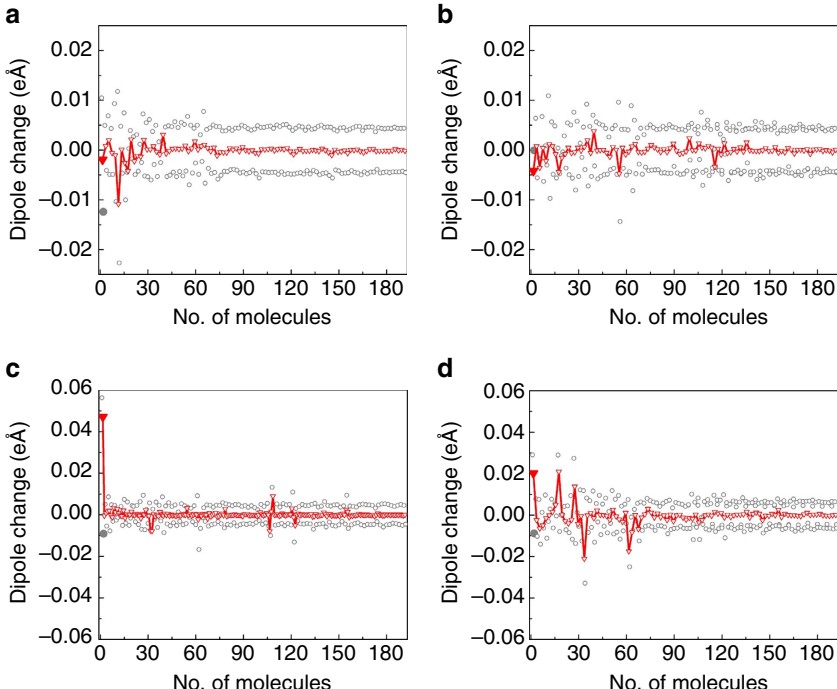

**Figure 4 | Dipole change of each molecule in the supercell.** Dipole change from 30 to 60 K of glycine doped with (**a**) L-alanine and (**b**) L-threonine. Dipole change of glycine doped with L-serine (**c**) from 30 to 60 K and (**d**) 100 to 130 K (note the different scales). The molecules are numbered from 1 to 192, where the dipole changes of the molecules are represented as empty grey circles and the dopant is marked as a filled grey circle. The molecules are sorted by distance from the dopant as D, L or D', L' pairs. The empty red triangles represent the total change in polarization of each pair, while the filled red triangle represents the dopant site (guest-glycine pair).

**Table 2 | Temperature dependence of the pyroelectric coefficient.**

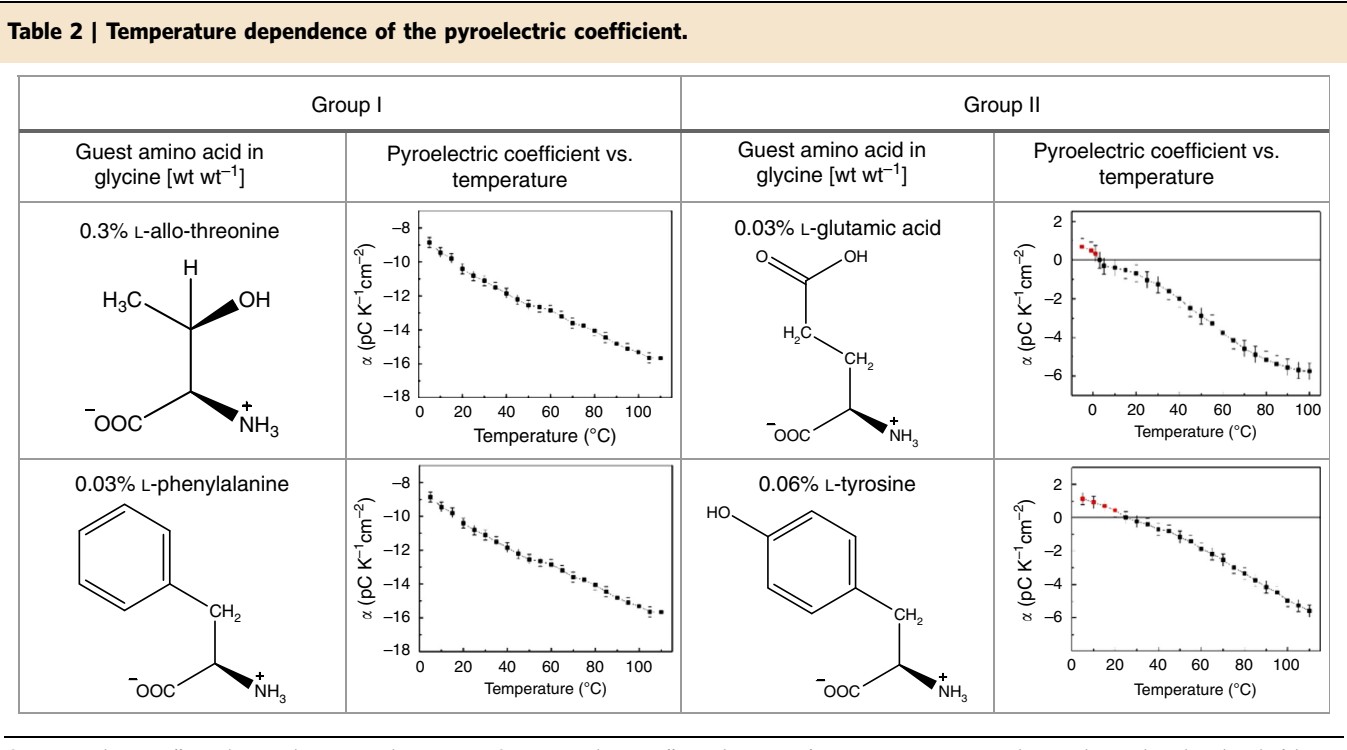

Group I: pyroelectric coefficient does not change sign with temperature; Group II: pyroelectric coefficient changes sign from positive to negative upon heating. The error bars show the s.d. of the pyroelectric coefficient at each temperature. Error bars represent s.e.m. values.

details). To provide additional support for the role that the acidic hydrogen of the dopant appears to play, thereby explaining the different behaviour of L-serine and L-threonine, we additionally examined experimentally the pyroelectricity of α-glycine doped with L-allo-threonine, L-phenylalanine, L-tyrosine and L-glutamic acid. The sign, magnitude and temperature dependence of the pyroelectric coefficient differ dramatically for various dopants, exhibiting two distinct types of behaviour (Table 2). Allo-threonine behaves similarly to threonine (Group I, negative pyroelectric coefficient within the range 5–110 °C), suggesting an intramolecular hydrogen bond. Glutamic acid and tyrosine (which bear an acidic hydrogen that can make an intermolecular H-bond) show a change in the sign of the pyroelectric coefficient as a function of temperature (Group II), similar to serine. Accordingly, phenylalanine (Group I), in contrast to tyrosine (Group II), does not form any side-chain group hydrogen bond, and the pyroelectric effect originates solely from the polarization induced by the distortion of the host due to the presence of the guest.

In conclusion, we have demonstrated that crystal engineering, pyroelectric measurements and neutron diffraction, together with judiciously constructed DFT and classical MD calculations, can be combined to determine the local structure of polar domains and their aggregate response, as induced by guest molecules, at concentrations as low as $<0.5\%$. Incorporating the experimental lattice parameters of the host, acquired at different temperatures, as a basis for the first-principles calculations expands their capability for predicting pyroelectric coefficients, and thereby, their ability to distinguish between possible conformations of the guest. The experimentally determined pyroelectric coefficient trends are in agreement with the calculated ones, demonstrating the reliability of the method. The differences between the pyroelectric responses of related molecules, such as L-serine and L-threonine, were shown to arise from the different conformations of the guest molecules within the host crystal. The ability to determine not only the conformation of the dopant,

but also the structure of the deformed host molecules in the vicinity of the guest site, should provide a rational methodology for the design of functional materials by doping, and for understanding the macroscopic polarity of crystals and related materials at the molecular level.

## Methods

**Crystal growth.** Mixed crystals of α-glycine (group $P2_1/n$) were grown by slow evaporation in a clean room environment at 23 °C from aqueous solutions of glycine (Alfa Aesar 99.5 + %) in the presence of: 5% wt wt$^{-1}$ L-alanine (Sigma ≥ 98%), L-threonine (T-Fisher Scientific 99.0–101.0%), L-allo-threonine (Alfa Aesar 99%), L-serine (Sigma ≥ 99%), 1.5% wt wt$^{-1}$ L-glutamic acid (Chem-Impex Int'l. Inc. + 99%), 0.1% wt wt$^{-1}$ L-phenylalanine (Sigma ≥ 98%) or L-tyrosine (Merck 99%). The mixed crystals were grown from a glycine solution in the presence of the chiral guest molecules, without any seeds being added to the solution. All commercial materials were used as received.

**Pyroelectric measurement.** The pyroelectric current of the mixed crystal was measured by the periodic temperature change technique[19] (Chynoweth method, see Supplementary Fig. 2a). The sample was heated by an infrared (IR) laser (3.5 W, $\lambda = 1.47$ μm wavelength) with a 2 W cm$^{-2}$ heat flux, which is transistor–transistor logic (TTL)-modulated by a DG4062 RIGOL waveform generator. The generated current was measured by a low impedance ($<10$ kΩ at $10^9$ V A$^{-1}$, $<500$ Ω at $10^8$ V A$^{-1}$) variable gain low noise current amplifier, DLPCA-200, and recorded with a digital averaging scope. The measurements were performed in a Faraday chamber having a slit for the laser beam and light absorbing inner coating. The bottom contact was prepared by fast drying silver paint and the top contact by carbon black conductive paint to ensure complete light absorption (5–50 μm thick). The pyroelectric coefficient as a function of temperature was measured by bringing the sample holder to the required temperature. The sample was kept at the required temperature for 15 min before the measurements. The measurement at each temperature was repeated at least eight times.

**Modelling using dispersion-corrected DFT.** The doped glycine crystal was modelled using a supercell containing $4 \times 2 \times 4$ unit cells of α-glycine, along the $a$, $b$ and $c$ lattice vectors of the α-glycine unit cell. This super cell contains 128 molecules, with one glycine molecule replaced by dopant. This leads to a doping concentration of 0.78%, which is of the same order of magnitude found in the experimental systems (0.3% wt wt$^{-1}$) and within the dilute limit, that is, with negligible inter-dopant interaction. At low dopant concentrations, the pyroelectric coefficient scales linearly, so the results of the calculations can be renormalized in

accordance with the actual dopant content. $P(T)$, the polarization along $b$ at temperature $T$ was calculated by setting the dimensions of the supercell to match the experimental unit cell of glycine at different temperatures: 5, 28, 58, 88, 127 and 154 °C (refs 22,23).

All DFT calculations were performed using the generalized-gradient-approximation exchange-correlation functional of Perdew, Burke and Ernzerhof[34], augmented by Tkatchenko–Scheffler dispersion-correction terms[35]. The Brillouin zone of the supercell was sampled with a single $k$-point[36] for the force relaxation calculations. For the Berry phase calculations, three $k$-points were used in the direction of the $b$ lattice vector, in order to attain convergence of the polarization component. The total energy was converged to $10^{-6}$ eV cell$^{-1}$ in all calculations and all forces in the optimized structures were smaller than 0.01 eV Å$^{-1}$. The calculations were performed in the accurate setting of Vienna *ab initio* simulation package (VASP)[37], a projector-augmented plane-wave code, with an energy plane-wave cutoff of 520 eV.

**Modelling using classical MD.** All classical MD simulations were based on the February 2016 version of the CHARMM force field[38]. Calculations were performed at fixed number of particles, pressure and temperature (NPT simulations, using a Nosé–Hoover thermostat with a thermal inertia $Ms = 500$ AMU). The time step was 0.6 fs. The MD geometry was based on the DFT-computed configuration of the dopant and its surroundings, and 'padded' with additional glycine molecules that increased the overall number of molecules in the supercell to 192 (corresponding to a 0.52% dopant concentration).

**Data availability.** The data that support the findings of this study are available from the corresponding author on reasonable request.

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

## Acknowledgements

The manuscript is dedicated to Prof Jack D. Dunitz. The authors thank Dr Isabelle Weissbuch (Weizmann Institute), Dr Ariel Biller (Weizmann Institute) and Prof Oded Hod (Tel Aviv Univ) for their helpful discussions and Naama Halevi (Dept. of Chemical Research Support, Weizmann Institute) for performing the high-pressure liquid chromatography analysis. The authors express their appreciation to the Israeli Science Foundation (226/13), Pearlman fellowship and the Nancy and Stephen Grand Research Center for Sensors and Security. This research is made possible in part by the historic generosity of the Harold Perlman Family. A.M.R. acknowledges the support of the US Office of Naval Research, under grant N00014-14-1-0761. The authors thank Charles L. Kane for his fruitful discussions concerning polarization.

## Author contributions

M.L., I.L., D.E. and E.M. designed the experimental part of the work. E.M. grew the crystals and performed the pyroelectric experiments. E.M., D.E., M.L. and I.L. participated in the interpretation of the pyroelectric results and in their discussion. I.A. and Y.Q. performed all DFT and MD calculations, respectively, reported in this article, analysed the data and participated actively in the discussions. A.M.R. and L.K. supervised the computational work, assisted the analysis of the data and participated actively in the discussions. All authors participated in the writing of the manuscript.

## Additional information

**Competing financial interests:** The authors declare no competing financial interests.

DOI: 10.1038/ncomms14597     **OPEN**

# Corrigendum: Origin and structure of polar domains in doped molecular crystals

E. Meirzadeh, I. Azuri, Y. Qi, D. Ehre, A.M. Rappe, M. Lahav, L. Kronik & I. Lubomirsky

*Nature Communications* 7:13351  doi: 10.1038/ncomms13351 (2016); Published 8 Nov 2016; Updated 6 Mar 2017

In Table 2 of this Article, the graph displaying 'Pyroelectric coefficient versus temperature' for 0.03% L-phenylalanine incorrectly replicates the graph above. The correct version of Table 2 appears below as Table 1.

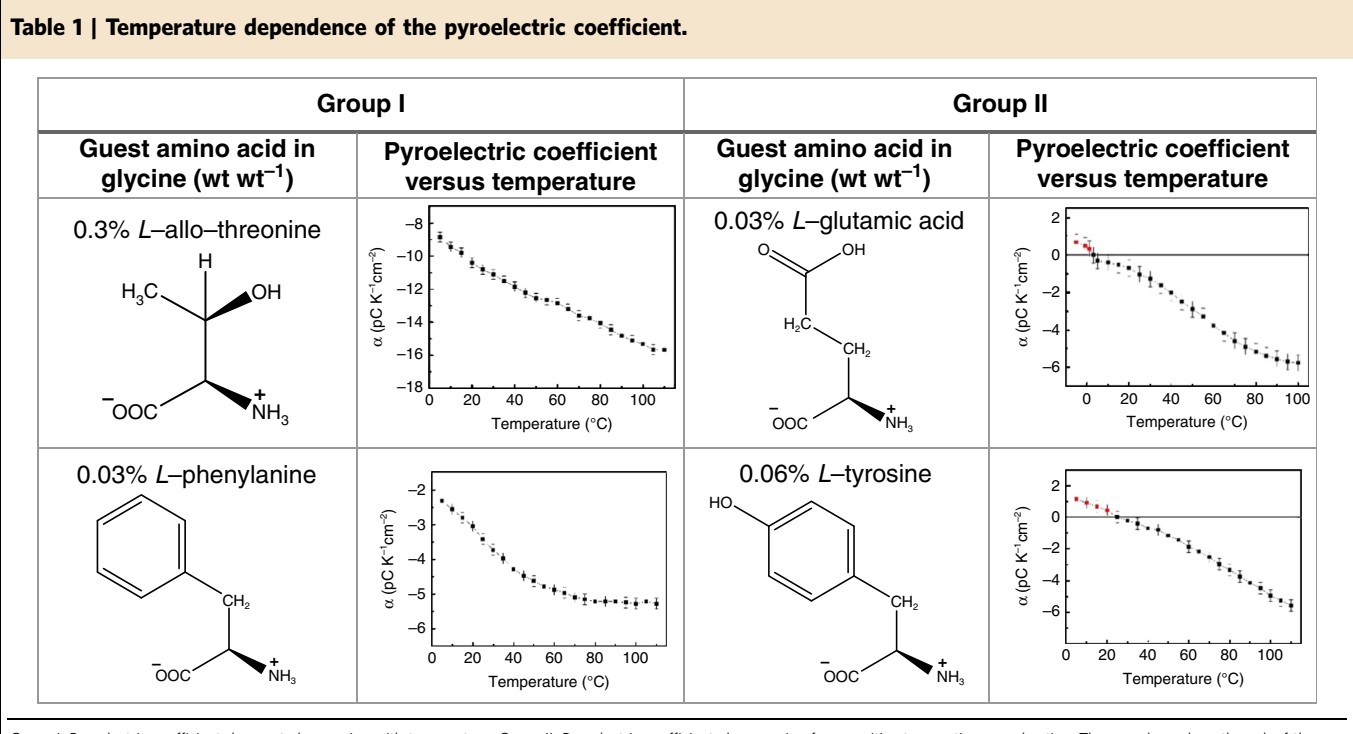

**Table 1 | Temperature dependence of the pyroelectric coefficient.**

| Group I | | Group II | |
|---|---|---|---|
| **Guest amino acid in glycine (wt wt⁻¹)** | **Pyroelectric coefficient versus temperature** | **Guest amino acid in glycine (wt wt⁻¹)** | **Pyroelectric coefficient versus temperature** |
| 0.3% *L*–allo–threonine | | 0.03% *L*–glutamic acid | |
| 0.03% *L*–phenylanine | | 0.06% *L*–tyrosine | |

Group I: Pyroelectric coefficient does not change sign with temperature; Group II: Pyroelectric coefficient changes sign from positive to negative upon heating. The error bars show the s.d. of the pyroelectric coefficient at each temperature. Error bars represent s.e.m. values.

DOI: 10.1038/ncomms15590 **OPEN**

# Erratum: Origin and structure of polar domains in doped molecular crystals

E. Meirzadeh, I. Azuri, Y. Qi, D. Ehre, A.M. Rappe, M. Lahav, L. Kronik & I. Lubomirsky

Nature Communications 7:13351 doi: 10.1038/ncomms13351 (2016); Published 8 Nov 2016; Updated 9 May 2017

In Fig. 1 of this Article, the top crystallographic face of the crystal was inadvertently mislabelled '$(0\bar{1}0)$' during the production process. It should read '(010)'. The correct version of Fig. 1 appears below.

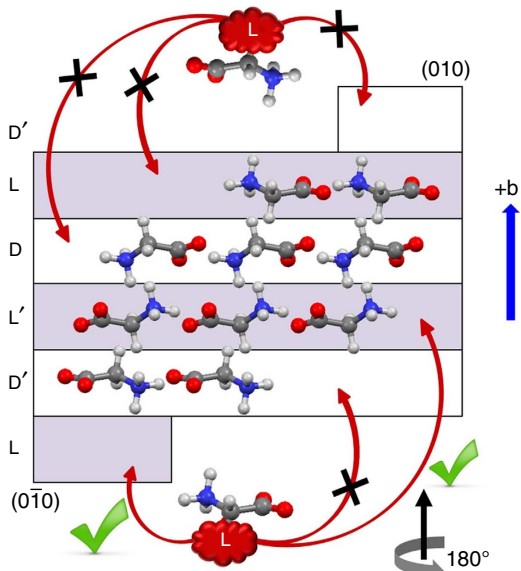

**Figure 1**

 1