## [Peer Review File · Nature Communications]

Reviewers' comments:

Reviewer #1 (Remarks to the Author):

The authors describe preparation, experimental characterization and theoretical description of alpha-glycine doped with L-amino acids. The dopant creates polar domains in the crystal which affect response of the whole crystal, making it polar. The structure is interpreted with combination of functional, structural and DFT studies.

It is in fact not clear to me what is the main purpose of this paper. It appears that it is, to quote authors, to "...demonstrate this integrated approach by determining the equilibrium molecular structure of the polar domains in centrosymmetric alpha-glycine crystals doped with different L-alpha-amino acids". I have the same impression from reading the paper. So, it appears that the paper is on a method rather than on specific results. This by itself is acceptable, if the method (a set of methods, in this case) brings something exceptional or is unusual. I am not sure this is the case for this paper.

I cannot comment on novelty of particular DFT approach used in this paper. In all other aspects, there are no particular novelties, in a qualitative sense. Or, if there are, this was not apparent to me.

The methods used are standard, the data appear to be of a good quality.

The main conclusion is about applicability of a set of experimental and theoretical techniques to determine the local structure of polar domains. The question that can then be posed is whether the same conclusion can be reached with another set of approaches or even with a single approach.

The paper is rather well written.

Reviewer #2 submitted remarks to the editor only. In these, it was suggested that the English phrasing in your paper would benefit from improvement for clarity.

Reviewer #3 (Remarks to the Author):

A. The manuscript NCOMMS-16-07813-T is devoted to the comprehensive experimental and theoretical studies of the polarization effects and related phenomena in the doped molecular crystals on the basis of glycine: alanine doped glycine, threonine doped glycine, and serine doped glycine. The peculiarities of interatomic interactions in the materials studied are presented in details. One of the strong parts of the manuscript are the combined experimental and theoretical studies of the polarization and related quantities, which have been shown to be in good agreement.

B. The results obtained in the manuscript are original.

C. The dispersion-corrected DFT-based calculations of the molecular crystals used ensure high degree of the agreement of the calculated results with corresponding experimental data.

D. The uncertainties have been taken into account appropriately.

E. In our opinion, the manuscript may be accepted for publication in Nature Communications after

minor corrections which follow.

F. Suggested improvements.

1. The Einstein relation for diffusion, $d = \sqrt{2Dt}$, should be used in the main manuscript instead of the relation $d = \sqrt{Dt}$ (line 93).
2. What are the details of the approach used for determination of the stable conformations of L-alanine, L-threonine, and L-serine in the host material? What was a criterion for the statements on the number of stable configurations in these doped materials studied? This should be added into the supplementary information of the manuscript.

G. References are with the appropriate credit to previous work.

H. The manuscript is written clear. The main parts of the manuscript (abstract, introduction, results and discussion, and conclusions) are in mutual accordance.

Reviewer #4 (Remarks to the Author):

This manuscript makes an excellent contribution to our understanding of the solid-state chemistry that explains the mode of incorporation of chiral molecules within achiral host crystals. Its great relevance is reflected both in that it addresses fundamental amino-acids as the system of study and, that it builds substantially on an outstanding body of published work covering a period of more than thirty years. A powerful combining of experimental and modelling approaches is presented building on (i) recent advances in the sensitivity of apparatus for measuring pyroelectric current in mixed crystal systems and (ii) a much improved treatment of dispersion forces within the formalism of fully periodic DFT calculations. Host crystals of alpha-glycine were grown by evaporation from aqueous solution in the presence of a number of L-amino acids particularly L-alanine, L-threonine and L-serine. The depth profile of the concentration of chiral guest molecule within a host crystal was determined by HPLC and inferred from time resolved measurements of the pyroelectric signal. The mode of incorporation of guest molecules in the host lattice was modelled using appropriate DFT calculations which were also used to calculate polarization contributions and pyroelectric coefficients.

I have a small number of questions and points for clarification concerning both the manuscript and the supplementary information.

(1) In Figure S1 (a) in the crystal of pure-host glycine, as indicated in the figure, the surface (010) has grown out of the crystal habit, whereas there is a (0 $\bar{1}$ 0) surface on the crystal, presumably because no mass transfer of solute was possible to the surface of the growing crystal that was directly in contact with the glass? How was it determined that the specific surface of the form {010} presented is the (0 $\bar{1}$ 0) surface and not the (010) surface? Similarly the shape of the doped crystal illustrated in figure S1 (b) is described as plate like (and therefore presumably equant) but is this as expected in the presence of a chiral impurity species?

(2) Can the authors further clarify how the crystals were grown in the presence of the chiral guest molecules for example, did the glycine crystals nucleate from a solution containing the additive or were seed crystals of pure host introduced to the doped solution and then grown? This relates to why a depth dependence of the concentration of the guest molecule is observed experimentally. If the surface area of the relevant growth sector presented to the solution during the growth process is taken into account, does this account for the observed depth profile of the guest molecule concentration in the crystal?

(3) In the dispersion corrected DFT calculations on a supercell of the host alpha-glycine crystal lattice with one host molecule substituted by a chiral guest molecule, as I understand it, only lattice positions with the same, inherent chirality, L or L' were considered in the point substitution. Could the authors comment on whether it is feasible for a chiral guest molecule to adopt an orientation and conformation at sites D or D' which would still accommodate the dipole of the single guest-molecule within the alignment of the dipoles of the host molecules? This would be interesting from the point of view of molecules site hopping within the lattice irrespective of whether the guest molecule initially had access to that site through a specific growth interface.

(4) Please could sets of Cartesian Coordinates, to describe the relative positions of all the atoms in the chiral guest-molecules for the conformations they adopt in a host glycine crystal (as calculated via DFT), be included in the supplementary material? This information would be very useful for comparison with the gas-phase minimum energy conformation as well as that found in the crystal structures of the guest molecules.

(5) In general I am struggling to see all the important information that is contained within the figures, particularly Figures 3 and 4. These will need to be enlarged and a higher resolution used. In Figure 3 the caption does not refer to the letters provided in the figure, this would improve the description.

I believe this excellent work should be published in Nature Communications but I would be grateful if the authors could address my questions and, if they think the points have merit, if they could add some additional text to the manuscript specifically relating to points (1) - (3).

Detailed response to the reviewer comments

Reviewer #1 (Remarks to the Author):

The authors describe preparation, experimental characterization and theoretical description of alpha-glycine doped with L-amino acids. The dopant creates polar domains in the crystal which affect response of the whole crystal, making it polar. The structure is interpreted with combination of functional, structural and DFT studies.

It is in fact not clear to me what is the main purpose of this paper. It appears that it is, to quote authors, to "...demonstrate this integrated approach by determining the equilibrium molecular structure of the polar domains in centrosymmetric alpha-glycine crystals doped with different L-alpha-amino acids". I have the same impression from reading the paper. So, it appears that the paper is on a method rather than on specific results. This by itself is acceptable, if the method (a set of methods, in this case) brings something exceptional or is unusual. I am not sure this is the case for this paper.

I cannot comment on novelty of particular DFT approach used in this paper. In all other aspects, there are no particular novelties, in a qualitative sense. Or, if there are, this was not apparent to me. The methods used are standard, the data appear to be of a good quality. The main conclusion is about applicability of a set of experimental and theoretical techniques to determine the local structure of polar domains.

The question that can then be posed is whether the same conclusion can be reached with another set of approaches or even with a single approach.

The paper is rather well written.

As a starting point we would like to reiterate that doping of organic (and inorganic) crystals is a foundation of modern materials science. From this point of view, understanding how dopants incorporate into molecular crystals is of great theoretical and practical importance. The reviewer is correct that the manuscript presents a method, however, this method provides unique capabilities. Until now, there was no way to determine the local environment of a dopant molecule, present in a small quantity in a molecular crystal. This was especially so if the dopant molecule had some similarities to the molecules of the host, as it is often the case. These difficulties stemmed from the absence of an analytical technique suitable for the task.

Our work is meant to demonstrate that understanding of the local structure can be achieved on a molecular and atomic level by combining a number of experimental techniques, with advanced pyroelectric measurements at the center, together with advanced DFT (and in the revised version, also MD) calculations. It is the conjunction of both theoretical and experimental parts of the work that we deem to be the most important. To further emphasize this claim, we added complementary MD simulations, which show the temperature dependence of polarization and the pyroelectricity of each molecule in the mixed crystals.

We chose doped glycine as a model system since it allows a comparison between different dopants within the same matrix. However, the obtained results are novel and very important since they provide a correlation between the macroscopic pyroelectricity and the microscopic molecular structure. Moreover, the results are exceptionally unusual because small differences in the dopant-host interaction, results in very different pyroelectric behavior. In addition, we have deciphered between the polarization of the dopant and the one it induces to the matrix, which could not be done by any experimental or theoretical technique taken separately.

Reviewer #2 submitted remarks to the editor only. In these, it was suggested that the English phrasing in your paper would benefit from improvement for clarity.

We made every effort to ascertain that the English in the manuscript is of the highest standard, especially as some of the authors are native speakers and/or write at a native speaker level. Nevertheless, we would very much appreciate it if specific comments to be brought to our attention, so that we may address them.

Reviewer #3 (Remarks to the Author):

A. The manuscript NCOMMS-16-07813-T is devoted to the comprehensive experimental and theoretical studies of the polarization effects and related phenomena in the doped molecular crystals on the basis of glycine: alanine doped glycine, threonine doped glycine, and serine doped glycine. The peculiarities of interatomic interactions in the materials studied are presented in details. One of the strong parts of the manuscript are the combined experimental and theoretical studies of the polarization and related quantities, which have been shown to be in good agreement.

B. The results obtained in the manuscript are original.

C. The dispersion-corrected DFT-based calculations of the molecular crystals used ensure high degree of the agreement of the calculated results with corresponding experimental data.

D. The uncertainties have been taken into account appropriately.

E. In our opinion, the manuscript may be accepted for publication in Nature Communications after minor corrections which follow.

We thank the referee very much for this assessment.

F. Suggested improvements.

1. The Einstein relation for diffusion, $d = \sqrt{2Dt}$, should be used in the main manuscript instead of the relation $d = \sqrt{Dt}$ (line 93).

Corrected accordingly.

2. What are the details of the approach used for determination of the stable conformations of L-alanine, L-threonine, and L-serine in the host material? What was a criterion for the statements on the number of stable configurations in these doped materials studied? This should be added into the supplementary information of the manuscript. Initial geometries of the guest molecules were chosen based on their possible open and closed conformations that can form a maximum number of hydrogen bonds. This point and the relevant references were incorporated in the main text.

G. References are with the appropriate credit to previous work.

H. The manuscript is written clear. The main parts of the manuscript (abstract, introduction, results and discussion, and conclusions) are in mutual accordance.

Reviewer #4 (Remarks to the Author): This manuscript makes an excellent contribution to our understanding of the solid-state chemistry that explains the mode of incorporation of chiral molecules within achiral host crystals. Its great relevance is reflected both in that it addresses fundamental amino-acids as the system of study and, that it builds substantially on an outstanding body of published work covering a period of more than thirty years. A powerful combining of experimental and modelling approaches is presented building on (i) recent advances in the sensitivity of apparatus for measuring pyroelectric current in mixed crystal systems and (ii) a much improved treatment of dispersion forces within the formalism of fully periodic DFT calculations. Host crystals of alpha-glycine were grown by evaporation from aqueous solution in the presence of a number of L-amino acids particularly L-alanine, L-threonine and L-serine. The depth profile of the concentration of chiral guest molecule within a host crystal was determined by HPLC and inferred from time resolved measurements of the pyroelectric signal. The mode of incorporation of guest molecules in the host lattice was modelled using appropriate DFT calculations which were also used to calculate polarization contributions and pyroelectric coefficients. I have a small number of questions and points for clarification concerning both the manuscript and the supplementary information.

We thank this reviewer for careful reading and understanding the manuscript and its implications.

(1) In Figure S1 (a) in the crystal of pure-host glycine, as indicated in the figure, the surface (010) has grown out of the crystal habit, whereas there is a (0 $\bar{1}$ 0) surface on the crystal, presumably because no mass transfer of solute was possible to the surface of the growing crystal that was directly in contact with the glass? How was it determined that the specific surface of the form {010} presented is the (0 $\bar{1}$ 0) surface and not the (010) surface? Similarly the shape of the doped crystal illustrated in figure S1 (b) is described as plate like (and therefore presumably equant) but is this as expected in the presence of a chiral impurity species?

The crystals of pure glycine may in principle expose either (010) or (0 $\bar{1}$ 0) faces at the crystal-glass interface. In the case of the crystal doped with an L-amino acid (Fig. S1 (b)), we know that the dopant is incorporated through the top (0 $\bar{1}$ 0) face because no mass transfer of solute was possible to the bottom surface. Following the reviewer comment and for the purpose of comparison between the pure crystal and the one doped with an L-amino acid, we changed the notation in the pure crystal from (0 $\bar{1}$ 0) to (010).

(2) Can the authors further clarify how the crystals were grown in the presence of the chiral guest molecules for example, did the glycine crystals nucleate from a solution containing the additive or were seed crystals of pure host introduced to the doped solution and then grown? This relates to why a depth dependence of the concentration of the guest molecule is observed experimentally. If the surface area of the relevant growth sector presented to the solution during the growth process is taken into account, does this account for the observed depth profile of the guest molecule concentration in the crystal?

No seeds were used during the growth of the crystals.

The reviewer is correct regarding the source of concentration gradient. We added this point to the manuscript.

(3) In the dispersion corrected DFT calculations on a supercell of the host alpha-glycine crystal lattice with one host molecule substituted by a chiral guest molecule, as I understand it, only lattice positions with the same, inherent chirality, L or L' were considered in the point substitution. Could the authors comment on whether it is feasible for a chiral guest molecule to adopt an orientation and conformation at sites D or D' which would still accommodate the dipole of the single guest-molecule within the alignment of the dipoles of the host molecules? This would be interesting from the point of view of molecules site hopping within the lattice irrespective of whether the guest molecule initially had access to that site through a specific growth interface.

Pyroelectric properties are fully reversible upon heating and cooling, which implies that no exchange of molecules between D and D' sites takes place. This fully agrees with a very large body of the previous studies on doped glycine.

(4) Please could sets of Cartesian Coordinates, to describe the relative positions of all the atoms in the chiral guest-molecules for the conformations they adopt in a host glycine crystal (as calculated via DFT), be included in the supplementary material? This information would be very useful for comparison with the gas-phase minimum energy conformation as well as that found in the crystal structures of the guest molecules

Gladly. A CIF file is included with the revision.

(5) In general I am struggling to see all the important information that is contained within the figures, particularly Figures 3 and 4. These will need to be enlarged and a higher resolution used. In Figure 3 the caption does not refer to the letters provided in the figure, this would improve the description.

We have introduced appropriate corrections.

I believe this excellent work should be published in Nature Communications but I would be grateful if the authors could address my questions and, if they think the points have merit, if they could add some additional text to the manuscript specifically relating to points (1) - (3).

REVIEWERS' COMMENTS:

Reviewer #1 submitted comments to only the editor, which stated that the reviewer found all revisions satisfactory.

Reviewer #3 (Remarks to the Author):

A. The manuscript NCOMMS-16-07813A is devoted to the comprehensive experimental and theoretical studies of the polarization effects and related phenomena in the doped molecular crystals on the basis of glycine: alanine doped glycine, threonine doped glycine, and serine doped glycine. The peculiarities of interatomic interactions in the materials studied are presented in details. One of the strong parts of the manuscript are the combined experimental and theoretical studies of the polarization and related quantities, which have been shown to be in good agreement.

B. The results obtained in the manuscript are original, interesting and useful for similar studies of other materials.

C. The dispersion-corrected DFT-based calculations of the molecular crystals used ensure high degree of the agreement of

the calculated results with corresponding experimental data. Additional results obtained by using the classical molecular dynamics method are generally in good agreement with those obtained by DFT-based calculations and with experimental data.

D. The uncertainties have been taken into account appropriately.

E. The present version of the manuscript may be accepted for publication in Nature Communications as is.

F. Suggested improvements. Please, insert into the supplement materials the atomic coordinates of all atoms in the chiral guest molecules for the conformations they adopt in a host glycine crystal, as was suggested by Reviewer #4.

G. References are with the appropriate credit to previous work.

H. The manuscript is written clear. The main parts of the manuscript (abstract, introduction, results and discussion, and conclusions) are in mutual accordance.

Reviewer #4 (Remarks to the Author):

I am satisfied that the points raised by the Reviewers have been adequately addressed by the authors and recommend publication of the revised article.

Response to referees:

REVIEWERS' COMMENTS:

Reviewer #1 submitted comments to only the editor, which stated that the reviewer found all revisions satisfactory.

Thanks to Reviewer #1.

Reviewer #3 (Remarks to the Author):

A. The manuscript NCOMMS-16-07813A is devoted to the comprehensive experimental and theoretical studies of the polarization effects and related phenomena in the doped molecular crystals on the basis of glycine: alanine doped glycine, threonine doped glycine, and serine doped glycine. The peculiarities of interatomic interactions in the materials studied are presented in details. One of the strong parts of the manuscript are the combined experimental and theoretical studies of the polarization and related quantities, which have been shown to be in good agreement.

B. The results obtained in the manuscript are original, interesting and useful for similar studies of other materials.

C. The dispersion-corrected DFT-based calculations of the molecular crystals used ensure high degree of the agreement of the calculated results with corresponding experimental data. Additional results obtained by using the classical molecular dynamics method are generally in good agreement with those obtained by DFT-based calculations and with experimental data.

D. The uncertainties have been taken into account appropriately.

E. The present version of the manuscript may be accepted for publication in Nature Communications as is.

F. Suggested improvements. Please, insert into the supplement materials the atomic coordinates of all atoms in the chiral guest molecules for the conformations they adopt in a host glycine crystal, as was suggested by Reviewer #4.

We have uploaded the cif. files with all the coordinates as "Supplementary Dataset" files.

G. References are with the appropriate credit to previous work.

H. The manuscript is written clear. The main parts of the manuscript (abstract, introduction, results and discussion, and conclusions) are in mutual accord.

We thank the referee very much for this assessment.

Reviewer #4 (Remarks to the Author):

I am satisfied that the points raised by the Reviewers have been adequately addressed by the authors and recommend publication of the revised article.

We thank the referee very much for his comment.